# Quantitative analysis of the impact of climate variability and human activities on grassland productivity of the Qilian Mountain National Park, China

**Qiang Li** [1]*, **Guoxing He**[2], **Degang Zhang**[2], **Xiaoni Liu**[2]

**1** State Key Laboratory of Plateau Ecology and Agriculture/College of Agriculture and Animal Husbandry, Qinghai University, Xining, 810016, China, **2** College of Grassland Science, Gansu Agricultural University, Lanzhou, P. R. China

* 1245524440@qq.com

**Data Availability Statement:** All relevant data are within the paper.

**Funding:** This study was supported by State Key Laboratory of Plateau Ecology and Agriculture,

## Abstract

To quantitatively analyze the impact of climate variability and human activities on grassland productivity of China's Qilian Mountain National Park, this study used Carnegic-Ames-Stanford Approach model (CASA) and Integrated Vegetation model improved by the Comprehensive and Sequential Classification System (CSCS) to assess the trends of grassland NPP from 2000 to 2015, the residual trend analysis method was used to quantify the impact of human activities and climate change on the grassland based on the NPP changes. The actual grassland NPP accumulation mainly occurred in June, July and August (autumn); the actual NPP showed a fluctuating upward trend with an average increase of 2.2 g C·m$^{-2}$ a$^{-1}$, while the potential NPP increase of 1.6 g C·m$^{-2}$ a$^{-1}$ and human-induced NPP decreased of 0.5 g C·m$^{-2}$ a$^{-1}$. The annual temperature showed a fluctuating upward trend with an average increase of 0.1°C 10a$^{-1}$, but annual precipitation showed a fluctuating upward trend with an average annual increase of 1.3 mm a$^{-1}$ from 2000 to 2015. The area and NPP of grassland degradation caused by climate variability was significantly greater than that caused by human activities and mainly distributed in the northwest and central regions, but area and NPP of grassland restored caused by human activities was significantly greater than that caused by climate variability and mainly distributed in the southeast regions. In conclusion, grassland in Qilian Mountain National Park showed a trend of degradation based on distribution area, but showed a trend of restoration based on actual NPP. Climate variability was the main cause of grassland degradation in the northwestern region of study area, and restoration of grassland in the eastern region was the result of the combined effects of human activities and climate variability. Under global climate change, the establishment of Qilian Mountain National Park was of great significance to the grassland's protection and the grasslands ecological restoration that have been affected by humans.

Qinghai university (2023-ZZ-05). And National Natural Science Foundation of China (31160475; 61401439). The funders had no role in study design, data collection and analysis, decision to publish, or preparation of the manuscript.

**Competing interests:** The authors have declared that no competing interests exist.

## Introduction

Climate change/variability and human activities have caused, to some degree, 49.3% of grassland degradation worldwide [1–3]. Understanding the interaction between climate change and human activities is important to investigating grassland dynamic changes and preventing grassland degradation [4,5]. Grassland net primary productivity (NPP), the rate of net carbon fixation through photosynthesis by grassland vegetation, directly reflects grassland productive capacity in the natural environment, and is an important indicator of growth dynamics [6–8]. NPP is susceptible to climate (e.g. rainfall) and human interference (e.g. overgrazing) [9,10]. NPP is an easy and reliable index to measure vegetation changes through traditional and remote sensing approaches at large-scale [4,5], and also has been used as an indicator to monitor and assess the relative impact of climate variability and human activities on dynamic changes in grassland ecosystems [11–13]. For instance, Nemani *et al.* [14] studied the relationship between global land NPP and climate factors, and found that, from 1982 to 1999 the change of global climate increased NPP by 6%. Yan et al. [15] observed significant increased ($1.66 \text{ g C m}^{-2} \text{ a}^{-1}$) in grassland NPP in northern of China from 2000 to 2015, with 64.9% of the grassland area with an increase and 35.1% with a decrease in NPP. The Qilian mountain National Park is located on the eastern edge of the China's Qinghai-Tibet Plateau, which offers important ecological service for water conservation and biodiversity maintenance in northwest China, and is one of the most fragile areas under global climate change [16,17]. Grassland is the predominant vegetation type in Qilian Mountain National Park, accounting for 37.4% of the total land area [7,18]. Therefore, assessing the role of climate variability and human activities on grassland NPP is important in investigating the vegetation dynamics and preventing grassland degradation in Qilian Mountain National Park.

The Comprehensive and Sequential Classification System (CSCS) [19–21], determining the grassland type belonged to the category of potential natural vegetation types [22]. That was, which is the climax vegetation type at the site without human interference characterizing with high stability and full mature [7]. The difference between the potential NPP and the actual NPP represents the change status of NPP caused by human activities. The precipitation and annual accumulated temperature are the main ecological factors used by the CSCS system in classifying grassland types [23–26]. And in Carnegic-Ames-Stanford Approach model (CASA) [27], precipitation and annual accumulated temperatureis commonly used in estimating actual NPP of global vegetation [6,8]. For instance, Zhang *et al.* [23] improved the CASA model based on CSCS and improved the accuracy of the model in estimating actual NPP for different grassland types and different scales [23,24]. This improvement allow the CASA model to use only relatively easily obtained annual cumulative temperature and humidity data as parameters, along with remote sensing data in estimating grassland NPP with and without human interferences [24].

The objectives of our research, using the improved CASA to analyze the changes in NPP in Qilian Mountain National Park from 2000 to 2015, quantifying the impact of human activities and climate change on grassland through comparing potential and actual grassland NPP. The results obtained from this study would be helpful in understanding grassland dynamic changes in the region in recent years with regard to the relative impacts of climate change and human activities, and in improving regional grassland management and protection measures.

## Materials and methods

### Study area

The Qilian Mountain National Park is located in the eastern Qinghai-Tibetan Plateau, China ($94°10'$-$103°04'$E, $35°50'$-$39°19'$N) (Fig 1). There are four vegetation zones in the region, namely

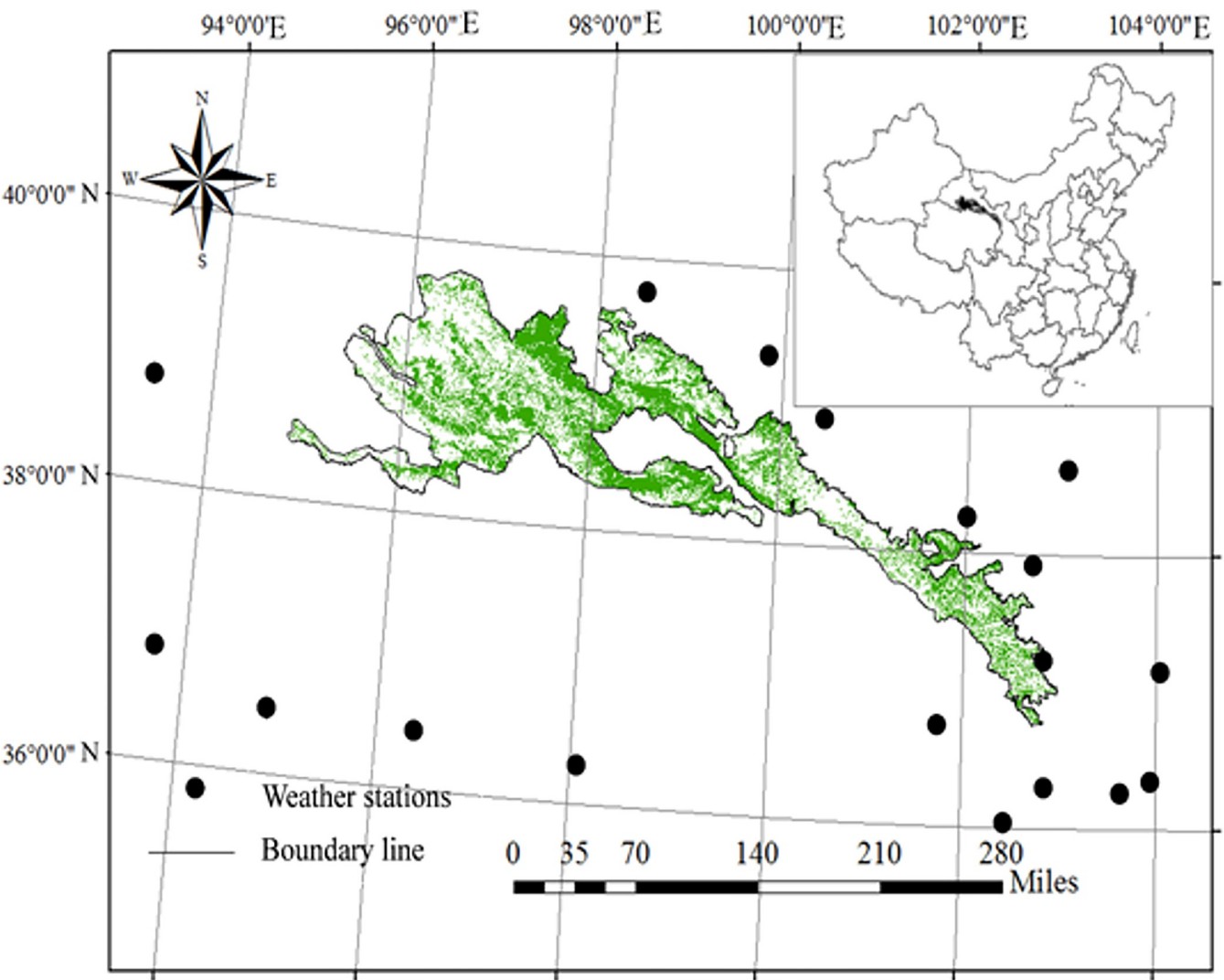

**Fig 1. Location of the research area.** The maps of Qilian Mountain National Park and China were downloaded from the Resource and Environment Science and Data Center Platform, China (DataV. GeoAtlas, https://www.resdc.cn) and using ArcGIS 10.2.2 platform split. Reprinted from (https://www.resdc.cn) under a CC BY license, with permission from Resource and Environment Science and Data Center Platform, original copyright 2019.

forest, shrub, grassland and desert extend in the order from southeast to northwest. Three vegetation belts, steppe, forest and alpine meadow, distributed vertically with increasing elevation (3000–5564 m). Major soil types include aridisols, inceptisols and entisols. Precipitation varies from 100 to 500 mm, occurring mostly from June to September. Average annual temperature is approximately -2.0°C; annual relative humidity ranges from 20% to 70%, and annual evaporation ranges from 1200 to1400 mm. The frost-free period is about 90–120 days [17].

## Data source and processing

### 1)The source of the Qilian Mountain National Park map

The maps of Qilian Mountain National Park and China (inspection number: GS Jing (2022) 1061) were downloaded from the Resource and Environment Science and Data Center Platform, China (DataV. GeoAtlas, https://www.resdc.cn) and using ArcGIS 10.2.2 platform split (ESRI, Redlands, CA: Environmental Systems Research Institute).

*2)Climate data*

Daily average temperature, monthly precipitation and monthly solar radiation data from 10 solar radiation stations and 19 temperature and precipitation stations for 2000 to 2015 were obtained from the China Meteorological Science Data Sharing Service Network (https://data.cma.cn/site/index.html) (Fig 1). Regression equations were developed between temperature, precipitation and solar radiation following the analytic method based on multiple regression and residues (AMMRR) developed by Guo, *et al.* [28] and Liu, *et al.* [29]. The I-AMMRR is an improved Ordinary Kriging interpolation using regression of climate data with altitude, slope and aspect [29]. Spatial data processing and analyses were conducted using ArcGIS 10.2.2 platform (ESRI, Redlands, CA: Environmental Systems Research Institute.).

*3) Remote sensing data*

NDVI data were derived from MODIS NDVI products sourced from the US Geological Survey (USGS) website for 2000 to 2015, with a temporal resolution of 16 days and spatial resolution of 1 km. The data were further processed through atmospheric, radiation and geometric correction and time-phase matching. The NDVI data of the first-half and the second-half of the month were then synthesized into monthly NDVI data, with a spatial resolution of 1 km.

*4) Other spatial data*

The measured actual NPP were sourced from the grassland survey data of Gansu Provincial Grassland Station in 2014 from102 sample points, including latitude and longitude, altitude, grassland type, and aboveground biomass. The DEM dataset was obtained from the Chinese geospatial data cloud (http://www.gscloud.cn/), with a spatial resolution of 90 m. Chinese administrative division data and the Qilian Mountain National Park boundary data were sourced from the Resource and Environmental Data Center of Chinese Academy of Sciences (http://www.resdc.cn/data.aspx?DATAID=282).

## Methods

**Estimation of actual NPP.** NPP is the product of absorbed photosynthetically active radiation (APAR) and the utilization efficiency (ξ) of vegetation on APAR that reaches the surface. It is expressed as:

$$NPP(x, t) = APAR(x, t) \times \xi(x, t)$$

where x is the spatial location, t is the time and ξ is light energy conversion rate. Photosynthetic effective radiation (APAR) depends on the proportion of total solar radiation and photosynthetic effective radiation absorbed by vegetation. NDVI data is used for calculation of APAR. The light energy conversion rate (ξ) refers to the efficiency of vegetation to convert the absorbed photosynthetically effective radiation into organic carbon, which is mainly affected by temperature and moisture, and expressed as:

$$\xi(x, t) = T_{\xi 1}(x, t) \times T_{\xi 2}(x, t) \times W_{\xi}(x, t) \times \xi_{MAX}$$

where $T_{\xi 1}(x, t)$ and $T_{\xi 2}(x, t)$ represent the effect of low and high temperature on the light energy conversion rate, $W_{\xi}(x, t)$ is a water stress factor, reflecting the effect of water content, $\xi_{MAX}$ is the maximum light energy utilization under ideal conditions.

Based on CSCS's cumulative temperature and humidity index, Zhang *et al.* [23] improved the CASA model by modifying the methods of calculating water stress factor and the

maximum light energy utilization rate in the light energy conversion. The improved water stress factor $W_\xi(x, t)$ was calculated by following equations [23]:

$$W_\xi(x, t) = 0.5 + 0.5 \times \frac{\left(0.29K^{\frac{1}{2}} + 0.6\right)\left(K \cdot L(K) + 0.469K^{\frac{3}{2}} + 9.33(\sum\theta)^{-1}\right)}{\left(K + 0.469K^{\frac{1}{2}} + 0.966\right)\left(L(K) + 0.933K^{-1}\right)}$$

$$L(K) = K + 0.906K^{\frac{-1}{2}} + 0.22$$

Where K is the humidity index of the grassland class [20] in the CSCS, which is calculated from the annual precipitation and cumulative temperature $\Sigma\theta > 0°C$:

$$K = \frac{P}{0.1\sum\theta}$$

**Estimate of potential NPP.** The basic formula for estimating potential NPP are as follows [25]:

$$NPP = L^2(k)\frac{0.1\sum\theta[k^6 + L(k)k^3 + L^2(k)]}{[k^6 + L^2(k)][k^5 + L(k)k^2]}e^{-\sqrt{13.55 + 3.17k^{-1} - 0.16k^2 + 0.0032k^{-3}}}$$

$$L(k) = 0.58802k^3 + 0.50698k2 - 0.0257081k + 0.0005163874$$

$$k = \frac{P}{0.1\sum\theta}$$

where P is annual precipitation, $\Sigma\theta$ is annual cumulative temperature $\Sigma\theta > 0°C$.

**Assessing the influence of climate and human activities on grassland productivity.** To quantitatively assess the impact of climate variability and human activities on grassland productivity, this study defined three forms of the NPP: the actual NPP ($N_a$), the potential NPP ($N_P$) and the human-induced NPP ($N_h$). The basic formula for calculating human-induced NPP ($N_h$) is as follows:

$$N_h = N_p - N_a$$

Distinguishing the beneficial or harmful effects of climate variability and human activities on grassland NPP can be achieved by calculating the slopes ($S_a$, $S_p$ and $S_h$) of the three types of NPPs [9,24,30]. The basic formula for calculating slopes is:

$$S_\theta = \frac{n \times \sum_{i=1}^{n} i \times NPP_i - \sum_{i=1}^{n} i \sum_{i=i}^{n} NPP_i}{n \times \sum_{i=1}^{n} i^2 - \left(\sum_{i=1}^{n} i\right)^2}$$

Where $\theta$ is for actual, potential or human-induced NPP, $NPP_i$ is the NPP in year i (g C m$^{-2}$ a$^{-1}$). i is the annual variable (e.g.i = 1, 2, 3, . . ., 16).

If $S_a > 0$, it was considered that grassland restoration has occurred and climate variability or (and) human activities may have improved grassland restoration. If $S_a < 0$ it was considered that grassland degradation has occurred and climate variability or (and) human activities may have caused grassland degradation. In order to assess the effects of grassland degradation and restoration by climate variability and human activities on grassland NPP in the Qilian Mountain National Park from 2000 to 2015, six situations were defined based on the values of $S_a$, $S_p$ and $S_h$ (Table 1).

**Table 1. Conditions that indicate the driving forces of vegetation dynamics.**

| Condition | $S_a$ | $S_p$ | $S_h$ | Driving Forces of Vegetation Dynamics |
|---|---|---|---|---|
| Condition1 | >0 | <0 | >0 | Vegetation restoration primarily caused by human factors |
| Condition2 | >0 | >0 | <0 | Vegetation restoration primarily caused by climate factors |
| Condition3 | >0 | >0 | >0 | Vegetation restoration primarily caused by climate-human factors |
| Condition4 | <0 | >0 | <0 | Vegetation degradation primarily caused by human factors |
| Condition5 | <0 | <0 | >0 | Vegetation degradation primarily caused by climate factors |
| Condition6 | <0 | <0 | <0 | Vegetation degradation primarily caused by climate-human factors |

$S_a > 0$ represents grassland vegetation restoration occurred while, $S_a < 0$ represents grassland vegetation degradation occurred. $S_p > 0$ represents climate variability that improved grassland vegetation restoration, while $S_p < 0$ represents climate variability that caused grassland vegetation degradation. $S_h > 0$ represents human activities that improved grassland vegetation restoration, while $S_h < 0$ represents human activities that caused grassland vegetation degradation. In order to assess the degraded and restored effects of climate variability and human activities on grassland NPP in Qilian mountains from 2010 to 2015, six conditions were defined (Table 1).

# Results

## Improved CASA model accuracy

To assess the improved CASA model accuracy, a correlation analysis between modelled and measured NPP was conducted. There was a strong linear correlation ($R^2 = 0.789$, P < 0.001), indicating that the grassland NPP was adequately estimated by the improved CASA model (Fig 2).

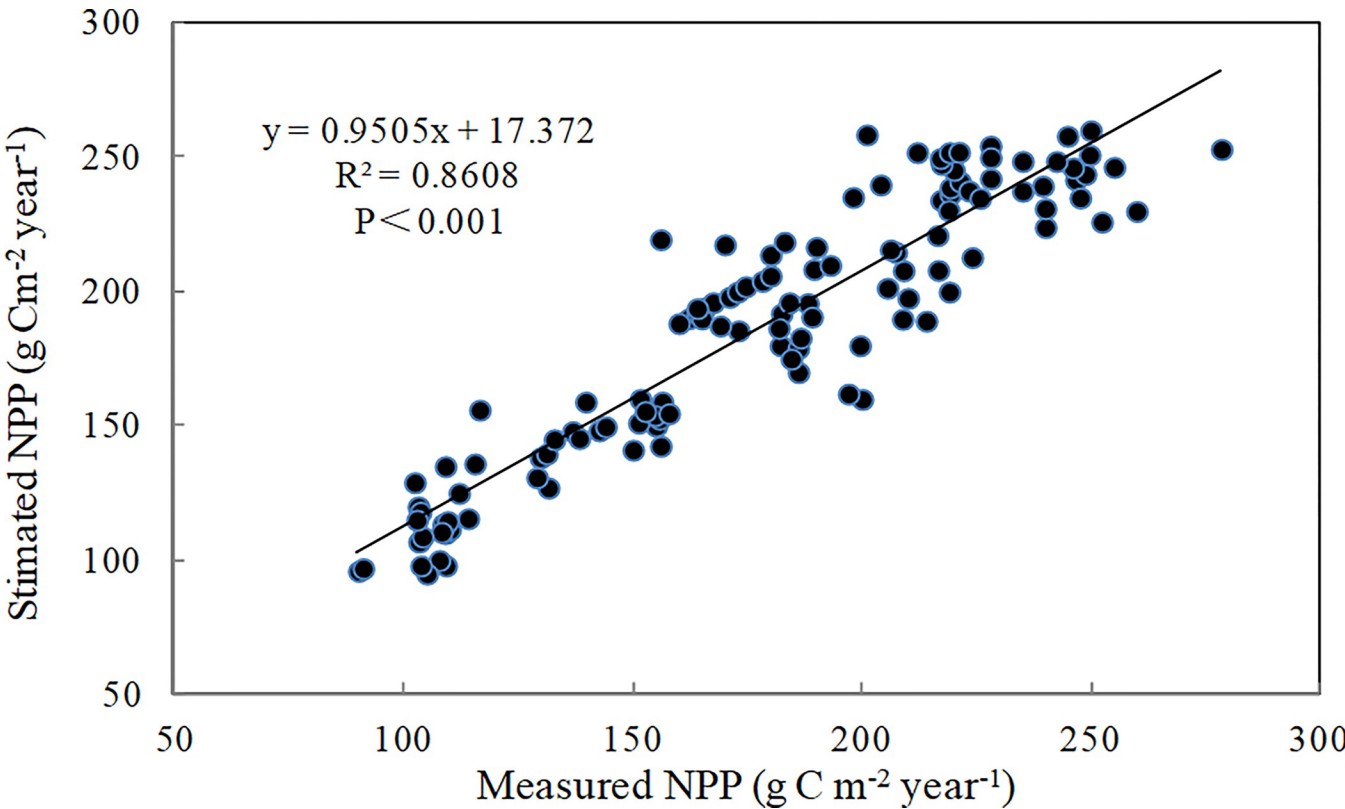

**Fig 2. Comparison of the estimated NPP with measured data.**

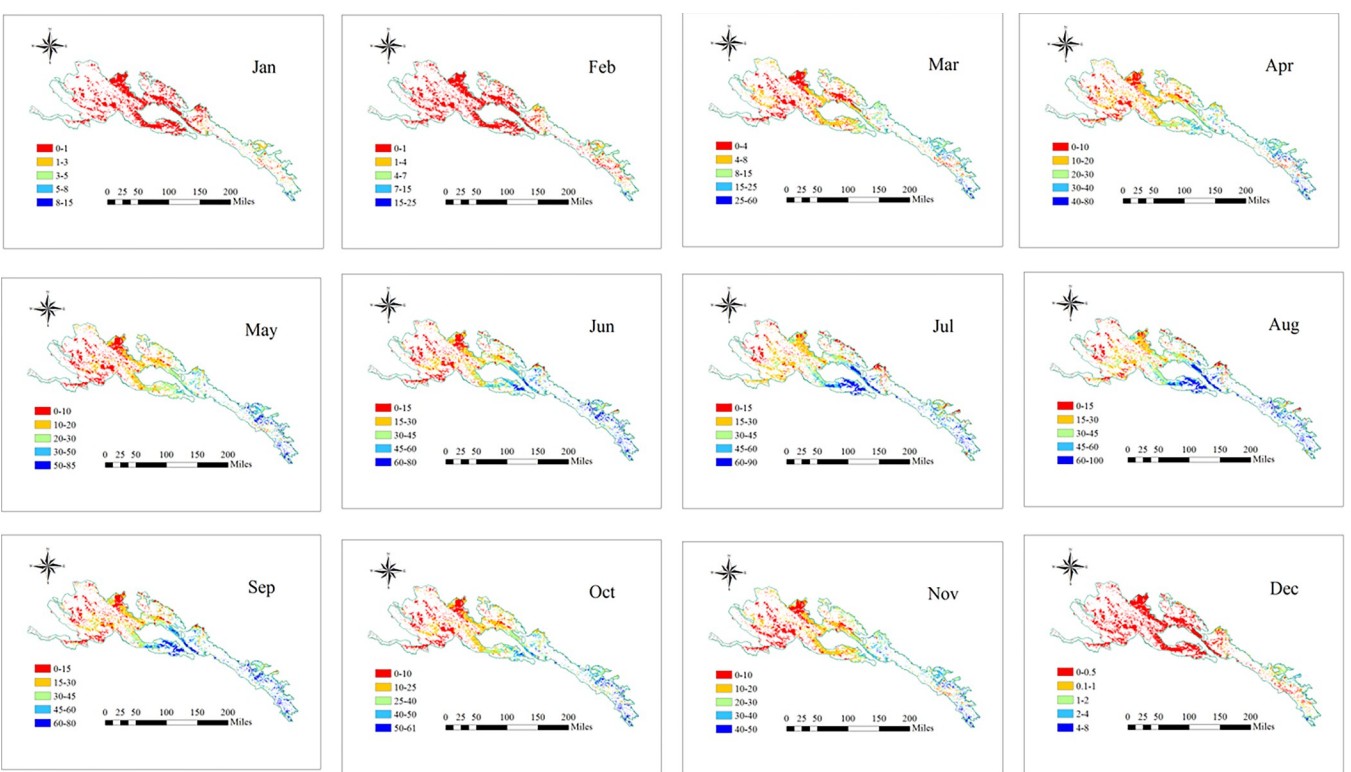

**Fig 3. Spatial distribution of the monthly grassland NPP in the Qilian mountains from 2010 to 2015.** The boundary was obtained from the Resource and Environment Science and Data Center Platform, China (DataV. GeoAtlas, https://www.resdc.cn).

## Actual NPP changes

In order to explore the month characteristics changes of NPP in 2000–2015, Calculate the average monthly NPP for 16 years in this research (Fig 3). From 2000 to 2015, The accumulation of grassland NPP *in* Qilian Mountain National Park grassland occurred mainly from April to October, accounting for 89.1% of total annual NPP (Fig 3). From January to February, NPP accumulation was low, but NPP accumulation start to increase in March. Maximum NPP accumulation occurred from June to August (Fig 5A). From September to December, grassland NPP accumulation decreased.

In order to explore the seasonality characteristics changes of NPP in 2000–2015, Calculate the average seasonality NPP for 16 years in this research (Fig 3). The NPP of the Qilian Mountain National Park was higher in the southeast and lower in northwest for all seasons from 2000 to 2015 (Fig 4). Spring, summer, autumn and winter NPP accounted for 3.3, 32.7, 51.9 and 12.1% of the total annual grassland NPP, respectively (Fig 5B).

From 2000 to 2015, the average annual actual NPP of the grassland showed a fluctuating upward trend (Fig 6), ranging from 141.4 to 195.4 g m$^{-2}$ a$^{-1}$. The average annual actual NPP was 163.8 g m$^{-2}$ a$^{-1}$, with an average annual increase of 2.2 g·C·m$^{-2}$Actual NPP showed an increasing trend from northwest to southeast (Fig 6).

## Potential and human-induced NPP changes

From 2000 to 2015, the average annual potential NPP of the grassland in Qilian Mountain National Park showed a fluctuating upward trend (Fig 7), ranging from 536.1 to 812.1 g m$^{-2}$ a$^{-1}$. The average annual NPP was 704.1 g m$^{-2}$ a$^{-1}$, with an average annual increase of 1.6 g C·m$^{-2}$.

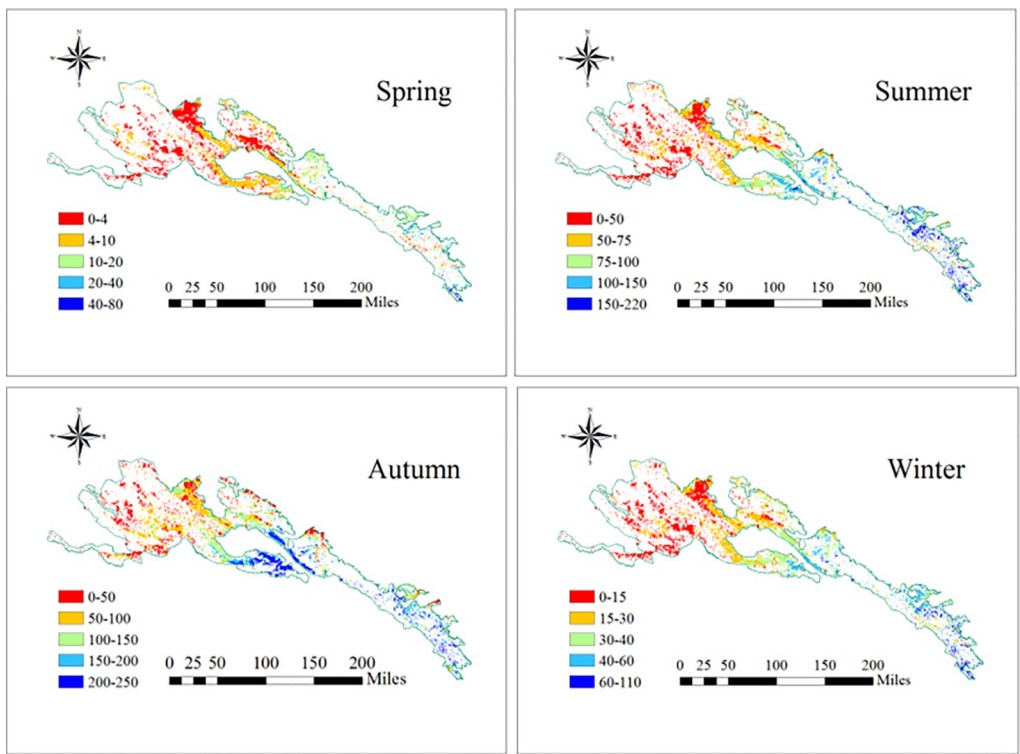

**Fig 4. Spatial distribution of the seasonal grassland NPP in the Qilian mountains from 2010 to 2015.** The boundary was obtained from the Resource and Environment Science and Data Center Platform, China (DataV. GeoAtlas, (https://www.resdc.cn).

The *human-induced NPP* showed a fluctuating downward trend (Fig 7), ranging from 340.7 to 648.8 g ($m^2$ a) −1. The average annual NPP was 540.3 g ($m^2$ a) −1, with an average annual decrease of 0.5 g C·$m^{-2}$.

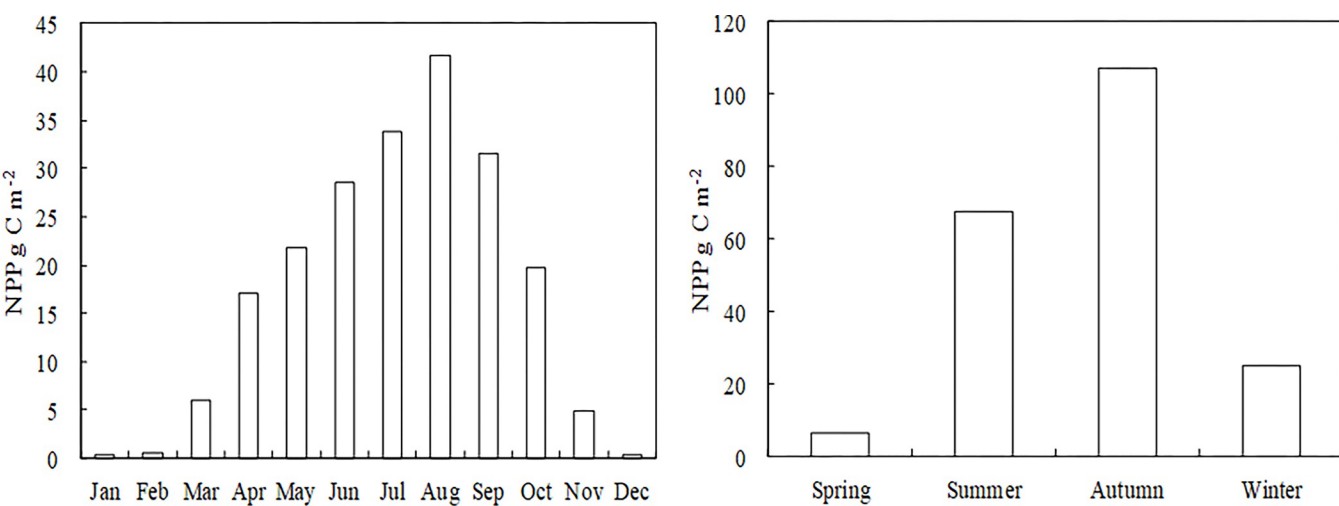

**Fig 5.** Spatial distribution of the average monthly (a) and seasonal (b) grassland NPP in the Qilian mountains from 2010 to 2015.

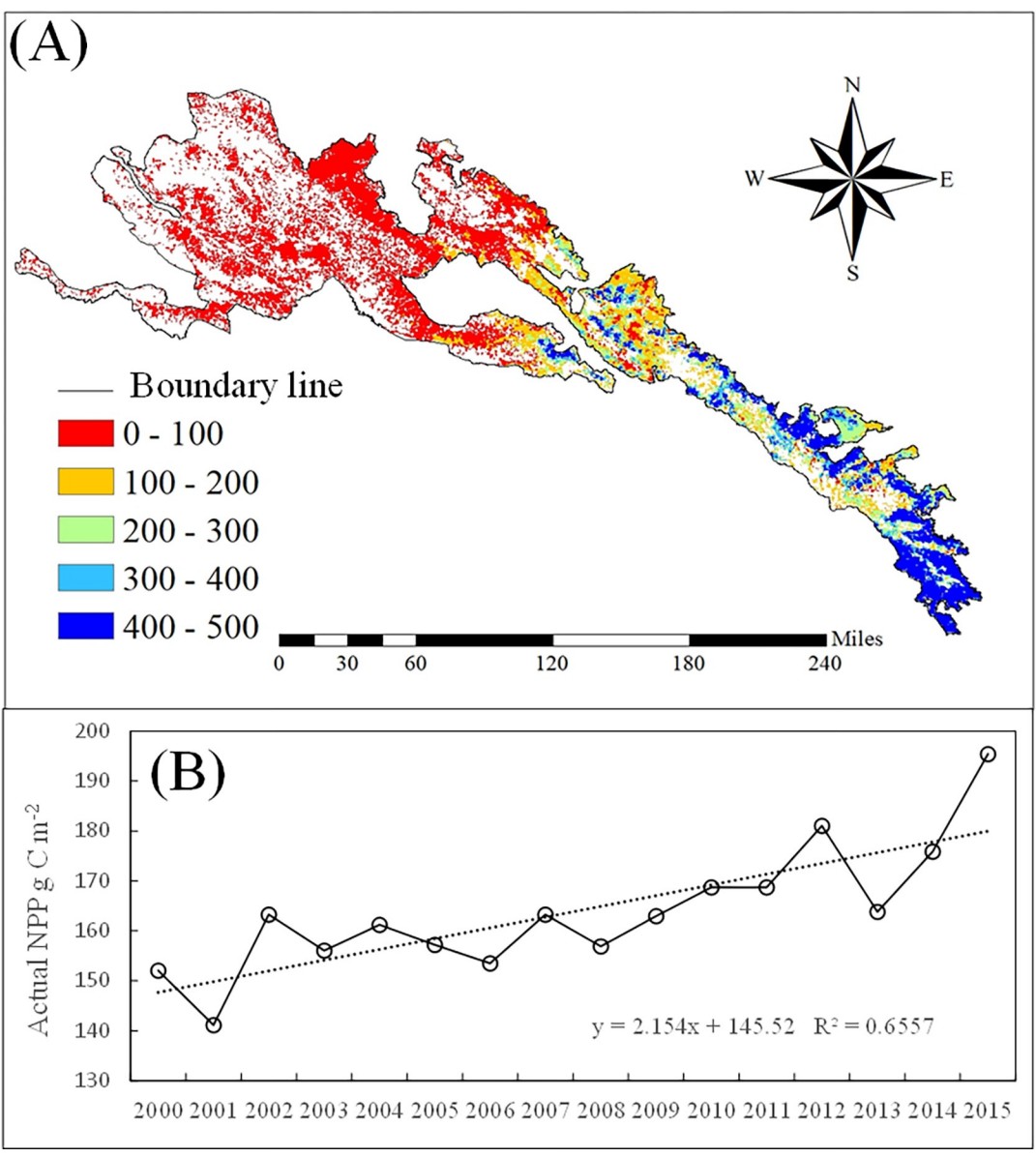

**Fig 6. Spatial distribution of the grassland Actual NPP in the Qilian mountain national park in 2000 to 2015.** The boundary was obtained from the Resource and Environment Science and Data Center Platform, China (DataV. GeoAtlas, https://www.resdc.cn).

### Effects of climate variability and human activities on grassland

Annual temperature (Fig 8A) and annual precipitation (Fig 8B) showed an increasing trend from northwest to southeast in Qilian Mountain National Park. From 2000 to 2015, the annual temperature showed a fluctuating upward trend (Fig 8C), ranging from -2.6 to -1.48°C. The average annual temperature was -2.1°C, with an average increase of 0.1°C 10a$^{-1}$. While average annual precipitation showed a fluctuating upward trend (Fig 8D), ranging from 319.8 to 519.8 mm. The average annual precipitation was 448.2mm, with an average increase of 1.3 mm a$^{-1}$.

The area of grassland degradation considered to be caused by climate variability and human activities was significantly larger than that of restoration (Fig 9A). The degraded

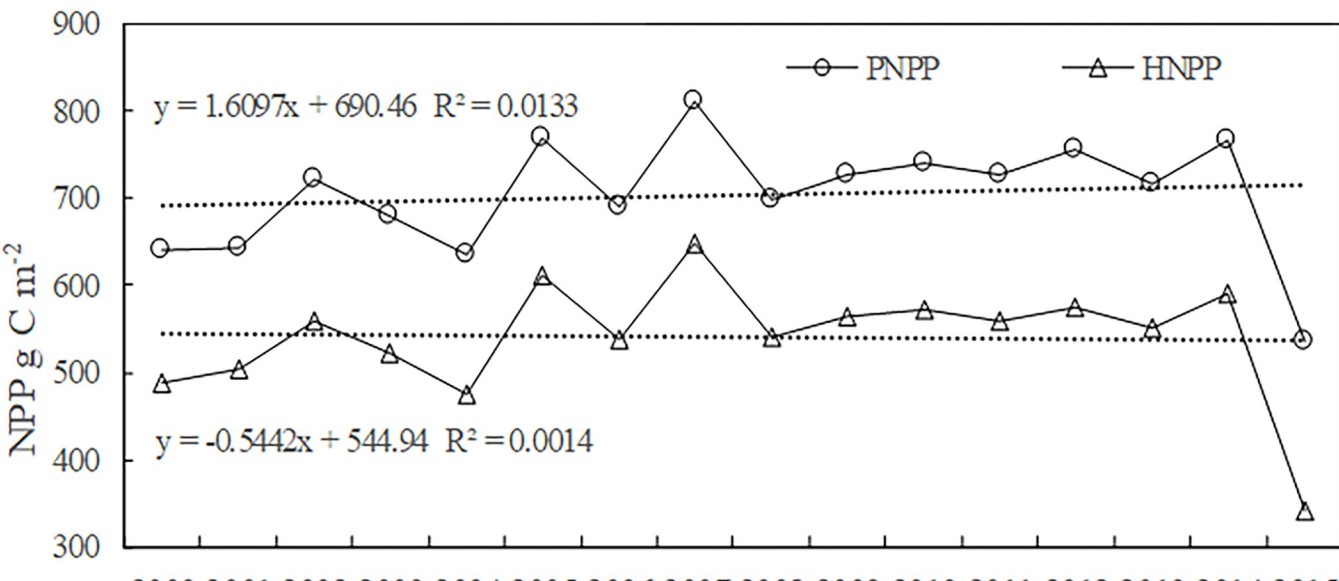

**Fig 7. Potential and human NPP changes in the Qilian mountain national park from 2000 to 2015.**

**Fig 8. The spatial distribution and change of annual precipitation and average temperature from 2000 to 2015.** The boundary was obtained from the Resource and Environment Science and Data Center Platform, China (DataV. GeoAtlas, https://www.resdc.cn).

grassland was mainly distributed in the northwest, while the restored grassland was mainly distributed in the southeast. Restored grassland considered to be caused by human activities, climate, climate-human activities accounted for 47.2%, 8.7% and 44.1% of the total restored grasslands area, respectively. Similarly, degraded grassland considered to be caused by climate and climate-human activities accounted for 96.9% and 3.1% of the total degraded grasslands area, respectively (Fig 9B and 9C). The influence of climate variability in grassland degradation and restoration was stronger than that of human activities. The area of degraded grassland caused by climate variability was the largest, accounting for 96.9% of the total degraded grassland area, which was mainly located in the northwestern.

Annual average actual NPP for grassland restored due to climate variability and human activities was significantly higher than that for degraded grassland (Fig 9D). The degraded grassland was mainly distributed in the northwest, while restored grassland was mainly distributed in the southeast. The NPP for restored grassland caused by human, climate, climate-human activities was 2207,1382 and 3130 Gg C·a $^{-1}$, respectively. While, the actual NPP for degraded grassland caused by climate and climate-human activities was 887 and 43 Gg C·a $^{-1}$, respectively (Fig 9E). The impact of climate variability on grassland degradation and restoration is higher than that of human activities.

## Discussion

The dynamic change of the grassland NPP is significant in measuring the health and adaptive mechanisms of regional grassland ecosystems [12,31], which also indicates the status of grassland performance under the combined effects of climate change and human activity. Previous studies showed that, the average NPP in the Qilian mountains varied from 204.92 to 336.8 g C·m$^{-2}$a$^{-1}$ during the 1985–2015, with overall NPP increasing from northwest to southeast [11,15]. Results from ours research showed that, from 2000 to 2015, the actual NPP of Qilian Mountain National Park showed a fluctuating upward trend, ranging from 141.4 to 195.4 g m$^{-2}$ a$^{-1}$, and the spatial distribution of NPP gradually decreased from east to west, the results were lower than those of Gang, *et al.* [11] and Yan *et al.* [15]. The mainly reason maybe that Qilian Mountain National Park is located in the ecologically fragile area of Qilian Mountain, which was reduced that the NPP was relatively small. Sun *et al.* [32] analyzed the spatial change characteristics of NPP in Qilian mountains from 2000 to 2010 based on MODIS data, and found that the vegetation NPP gradually decreased from east to west, which is consistent with the result from this study. There were obvious spatial differences in grassland NPP between the east and the west, which may be related to the regional topography, grassland type, soil type and other conditions [32,33]. In ours research, the low value of actual NPP appeared in 2001, and the peak value appeared in 2015. The 2001 was another year of severe droughts in China after the successive severe droughts in 1999 and 2000, the wind and sandstorms appeared early and frequently, especially in the central and western regions of Gansu Province, which had a significant impact on agriculture and animal husbandry production, transportation and people's lives. Meanwhile, aggravated the further deterioration of the regional ecological environment and seriously affects the growth of regional grassland. In 2015, precipitation in many parts of China increased by 20–100% and the temperature was higher, including the northwestern part of the Qinghai-Tibet Plateau, which made the climate tend to warm-humid, and conducive to the growth of grassland [32]. In the past 10 years, the temperature in the Qilian mountains generally increased, while precipitation increased in the east and decreased in central and western regions, with the increased temperature having a negative impact on grassland productivity [34,35]. Rong [36] also indicated that the average rate of temperature increased from 1957 to 2017 (0.32˚C/10a). Our results show that, from 2000 to 2015,

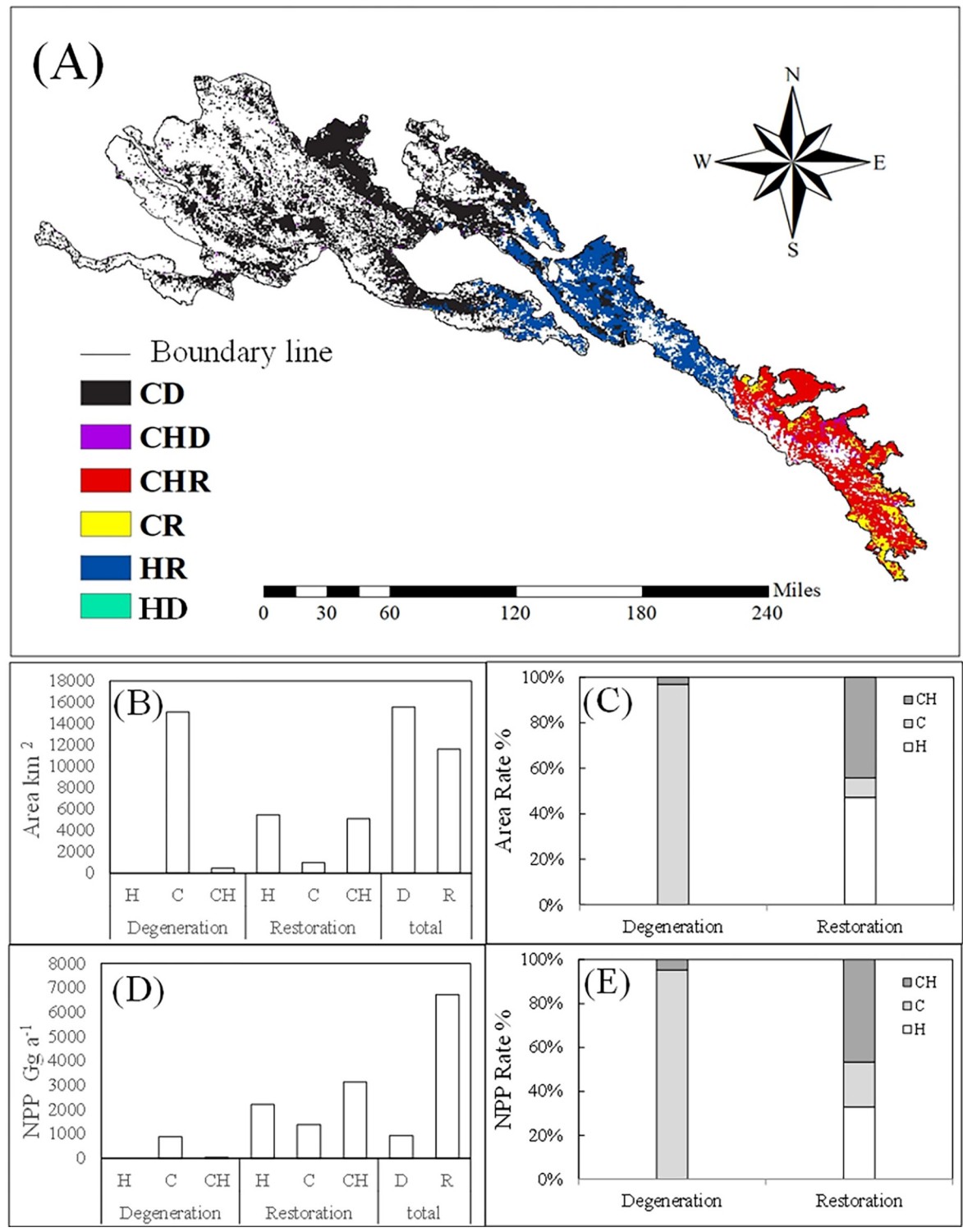

**Fig 9. Effects of climate variability and human activities on grassland NPP and area change.** HD: grassland degratation caused by human activities, HR: grassland restoration attributted by human activities, CD: grassland degratation caused by climate variability, CR: grassland restoration attributed by climate variability, CHD: grassland degratation caused by climate variability and human activities. The boundary was obtained from the Resource and Environment Science and Data Center Platform, China (DataV. GeoAtlas, https://www. resdc.cn).

the annual temperature showed a fluctuating upward trend with an average increase of 0.1˚C/ 10a[-1], but average annual precipitation showed a fluctuating upward trend with an average increase of 1.3 mm a[-1], spatial distribution of annual precipitation showed an increasing trend from northwest to southeast, indicating that the decreased precipitation and the increased temperature in northwest was the major factors causing decreased of grassland NPP. In other words, with the increased annual precipitation and decreased cumulative temperature from northwest to southeast in the Qilian Mountain National Park, the NPP increased accordingly.

Climate variability and human activities were the main factors affecting grassland changes [9,24,30], while human activities can strengthen or mitigate the influences of climate variability on ecosystems [10,37,38]. Yan *et al.* [15] and Zhu *et al.* [39] indicated that, the contribution of climate variability to grassland NPP changes was much greater than that of human activities in northern China. Our results show that, grassland degradation (NPP decreased) caused by climate variability was significantly greater than that caused by human activities, but grassland restoration (NPP increased) caused by human activities was significantly greater than that caused by climate variability, the region of grassland degradation was mainly located in the northwestern and region of grassland restoration was mainly located in the southeastern, which was similar to the results obtained by Li [7] and Rong [36]. Because of the Qilian Mountain National Park belongs to the transition zone of the Qinghai-Tibet Plateau, adjacent to the Loess Plateau and the Inner Mongolia Plateau, the impact of climate change/climate variability on grassland under such a fragile ecological environment is extremely significant. Climate variability was the main factor leading to the grassland degradation, followed by climate-human interaction. The impact of human activities on grassland changes was mainly in the east, because the population density distribution was relatively high, but was low in the central and sparse in the western parts of the Qilian mountians [40]. With the recent implementation of grassland ecological protection and residents migration projects, grazing pressure and population declined. Due to an improvement in herdsmens' awareness to ecological protection, the damage by human activities on grassland also declined [40,41]. Under global climate change, in view of the actual situation of grassland degradation in the Qilian Mountains, the state established the Qilian Mountain National Park to protect the grasslands in the fragile ecological environment.

Based on the CSCS theory constructed model to estimate the potential NPP and CASA model to estimate the actual NPP, and quantitatively described the impacts of human activities and climate change on the grassland NPP in Qilian Mountain National Park during 2000– 2015. Because the CASA model is a light energy use efficiency model and cannot predict future NPP changes, our research only analyzed the effects of human activities and climate change on grassland NPP in Qilian Mountain National Park during the period of 2000–2015, while lacked future projections. The use of CSCS theory to construct models and CASA models can quantitatively analyze climate change and human activities, which could provide a way of estimation for the evaluation of later ecological restoration projects and regional ecological evaluation.

## Conclusion

From 2000 to 2015, the actual NPP and potential NPP of the grassland in Qilian Mountain National Park showed a fluctuating upward trend, but human-induced NPP showed a fluctuating downward trend. The spatial distribution of NPP showed an increasing trend from northwest to southeast. the annual temperature and average annual precipitation showed a fluctuating upward trend with 0.1˚C 10a[-1] and 1.3 mm a[-1]. The grassland was degraded according to distribution area, but restored according to actual NPP. Climate variability was

the main cause of grassland degradation in the northwestern region of Qilian Mountain National Park, and restoration of grassland in the eastern region was the result of the combined effects of human activities and climate variability.

Under global climate change, the establishment of Qilian Mountain National Park was of great significance to the grassland's protection and the grasslands ecological restoration that have been affected by humans. Meanwhile, the assessment of grassland productivity in Qilian Mountain National Park is not only important for the ecology and economy of Qilian Mountain, but also for the ecology of the entire Qinghai-Tibetan Plateau and even the world's third pole region. Therefore, ours results of this study provide a basis for the scientific management (protection and restoration) of grassland in Qilian Mountain National Park and Qinghai-Tibetan Plateau.

## Supporting information

**S1 Fig.**
(PNG)

## Acknowledgments

The authors are grateful to the faculty members, students and technicians who collected the initial data, and provided guidance and assistance.

## Author Contributions

**Conceptualization:** Qiang Li.

**Data curation:** Qiang Li, Guoxing He, Xiaoni Liu.

**Formal analysis:** Qiang Li, Guoxing He.

**Funding acquisition:** Qiang Li, Xiaoni Liu.

**Writing – original draft:** Qiang Li.

**Writing – review & editing:** Qiang Li, Guoxing He, Degang Zhang, Xiaoni Liu.

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
