## [Decision Letter · Decision Letter 0]

11 Jul 2023

PONE-D-23-17536Quantitative analysis of the impact of climate variability and human activities on grassland productivity of the Qilian Mountain National Park, ChinaPLOS ONE

Dear Dr. Li,

Thank you for submitting your manuscript to PLOS ONE. After careful consideration, we feel that it has merit but does not fully meet PLOS ONE’s publication criteria as it currently stands. Therefore, we invite you to submit a revised version of the manuscript that addresses the points raised during the review process.

We look forward to receiving your revised manuscript.

Kind regards,

Tunira Bhadauria, Ph.D.

Academic Editor

PLOS ONE

Journal Requirements:

   "no"

6. We note that Figures 1,3 and 8 in your submission contain map/satellite images which may be copyrighted. All PLOS content is published under the Creative Commons Attribution License (CC BY 4.0), which means that the manuscript, images, and Supporting Information files will be freely available online, and any third party is permitted to access, download, copy, distribute, and use these materials in any way, even commercially, with proper attribution. For these reasons, we cannot publish previously copyrighted maps or satellite images created using proprietary data, such as Google software (Google Maps, Street View, and Earth). For more information, see our copyright guidelines: http://journals.plos.org/plosone/s/licenses-and-copyright.

a. You may seek permission from the original copyright holder of Figures 1,3 and 8 to publish the content specifically under the CC BY 4.0 license.  

Reviewers' comments:

Reviewer's Responses to Questions

**Comments to the Author**

1. Is the manuscript technically sound, and do the data support the conclusions?

Reviewer #1: Yes

Reviewer #2: Yes

2. Has the statistical analysis been performed appropriately and rigorously? 

Reviewer #1: Yes

Reviewer #2: N/A

3. Have the authors made all data underlying the findings in their manuscript fully available?

Reviewer #1: Yes

Reviewer #2: Yes

4. Is the manuscript presented in an intelligible fashion and written in standard English?

Reviewer #1: Yes

Reviewer #2: Yes

5. Review Comments to the Author

Reviewer #1: Manuscript entitled "Quantitative analysis of the impact of climate variability and human activities on

grassland productivity of the Qilian Mountain National Park, China" are recommended for publication. Although following modification are required

1.Future perspective must be included in conclusion

2.An abstract figure which may summaries whole work may be drawn

Reviewer #2: Dear Author, The manuscript Quantitative analysis of the impact of climate variability and human activities on

grassland productivity of the Qilian Mountain National Park, China has been written well and looks sound. However, some modifications are needed before publishing the MS. Introduction part is bit large so i would suggest to write it in more crisp way to attract the readers. In results section, figure 3 4 and figure 6 A, it is difficult to understand. For example figure 3, the Spatial distribution of the monthly grassland NPP has been shown there from January to December but the year 2000 to 2015 is not reflecting in the figure. Kindly look into this and explain if it is okay or make corrections if needed. The number of figures are too much in the MS, therefore, shift some figures in the supplementary figures. Check the fig. 6 where it is mentioned that 2000 and 2015, it must be 2000 to 2015. The discussion is still not that much supportive to your findings. Better to rewrite it giving some more conclusive remarks by supporting your findings. At this stage its like review of literature.

After these small corrections, the manuscript is recommended for publication.

6. PLOS authors have the option to publish the peer review history of their article (what does this mean?). If published, this will include your full peer review and any attached files.

Reviewer #1: **Yes: **Kuldip Jayaswall

Reviewer #2: No

<quillbot-extension-portal></quillbot-extension-portal>

---

## [Author Response · Author response to Decision Letter 0]

28 Jan 2024

RESPONSES TO EDITORS AND REVIEWERS

Aug 22, 2023

Tunira Bhadauria, Ph.D. 

Academic Editor 

PLOS ONE 

Dear Dr. Tunira Bhadauria, Ph.D.:

Thank you very much for your email and for the reviewers” Quantitative analysis of the impact of climate variability and human activities on grassland productivity of the Qilian Mountain National Park, China (PONE-D-23-17536)”. The comments were all valuable and very helpful in revising and improving our paper; these comments are also important guidance that is significant to our research. We studied the comments carefully and made corrections that we hope meet with your approval. We tried our best to improve the manuscript and made changes to the manuscript accordingly. These changes do not influence the content or framework of the paper. Once again, thank you very much for your comments and suggestions. 

The main corrections in the paper and the responses to the reviewer comments are as follows:

Response to Journal Requirements:

Response: we had amended.

Response: we had amended.

Response: we had amended.

4. Thank you for stating the following financial disclosure: "no"

Response: we had amended

5. In your Data Availability statement, you have not specified where the minimal data set underlying the results described in your manuscript can be found. PLOS defines a study's minimal data set as the underlying data used to reach the conclusions drawn in the manuscript and any additional data required to replicate the reported study findings in their entirety. All PLOS journals require that the minimal data set be made fully available. For more information about our data policy, please see http://journals.plos.org/plosone/s/data-availability. "Upon re-submitting your revised manuscript, please upload your study’s minimal underlying data set as either Supporting Information files or to a stable, public repository and include the relevant URLs, DOIs, or accession numbers within your revised cover letter. For a list of acceptable repositories, please see http://journals.plos.org/plosone/s/data-availability# loc-recommended-repositories. Any potentially identifying patient information must be fully anonymized.

Important: If there are ethical or legal restrictions to sharing your data publicly, please explain these restrictions in detail. Please see our guidelines for more information on what we consider unacceptable restrictions to publicly sharing data: http://journals.plos.org/plosone/s/data-availability#loc-unacceptable -data-access-restrictions. Note that it is not acceptable for the authors to be the sole named individuals responsible for ensuring data access.

Response: we had amended

6. We note that Figures 1,3 and 8 in your submission contain map/satellite images which may be copyrighted. All PLOS content is published under the Creative Commons Attribution License (CC BY 4.0), which means that the manuscript, images, and Supporting Information files will be freely available online, and any third party is permitted to access, download, copy, distribute, and use these materials in any way, even commercially, with proper attribution. For these reasons, we cannot publish previously copyrighted maps or satellite images created using proprietary data, such as Google software (Google Maps, Street View, and Earth). For more information, see our copyright guidelines: http://journals.plos.org/plosone/s/licenses-and-copyright.

 a. You may seek permission from the original copyright holder of Figures 1,3 and 8 to publish the content specifically under the CC BY 4.0 license. 

Response: we had amended

Response to Editor comments (Tunira Bhadauria, Ph.D):

Response: we had provided to explain your answers to the questions above.

Response to Reviewer #1 

Manuscript entitled "Quantitative analysis of the impact of climate variability and human activities on

grassland productivity of the Qilian Mountain National Park, China" are recommended for publication. Although following modification are required

1.Future perspective must be included in conclusion

Response: we had amended.

2.An abstract figure which may summaries whole work may be drawn

Response: we had amended.

Response to Reviewer #2:

Dear Author, The manuscript Quantitative analysis of the impact of climate variability and human activities on

grassland productivity of the Qilian Mountain National Park, China has been written well and looks sound. However, some modifications are needed before publishing the MS.

Introduction part is bit large so i would suggest to write it in more crisp way to attract the readers. 

Response: we had amended

In results section, figure 3 4 and figure 6 A, it is difficult to understand. For example figure 3, the Spatial distribution of the monthly grassland NPP has been shown there from January to December but the year 2000 to 2015 is not reflecting in the figure. Kindly look into this and explain if it is okay or make corrections if needed.

Response: we had amended.

 The number of figures are too much in the MS, therefore, shift some figures in the supplementary figures. Check the fig. 6 where it is mentioned that 2000 and 2015, it must be 2000 to 2015. 

Response: we had amended.

The discussion is still not that much supportive to your findings. 

Response: we had amended.

Better to rewrite it giving some more conclusive remarks by supporting your findings. At this stage its like review of literature.

Response: we had amended.

After these small corrections, the manuscript is recommended for publication.

In all we feel that these results are interesting and that all the parts to a focused manuscript are in the authors possession, they just need to refocus the paper and make sure that the purpose of the study, and discussion points are all clearly linked together and the results are put into a broader context.

We tried our best to improve the manuscript and made changes to the manuscript accordingly. These changes do not influence the content or framework of the paper. We appreciate the Editors’ and Reviewers’ earnest help, and hope that the corrections are sufficient. Once again, thank you very much for your comments and suggestions.

Yours sincerely,

Qiang Li

Corresponding author

E-mail: 1245524440@qq.com

---

## [Editor Report · Decision Letter 1]

1 Mar 2024

Quantitative analysis of the impact of climate variability and human activities on grassland productivity of the Qilian Mountain National Park, China

PONE-D-23-17536R1

Dear Dr. Li

We’re pleased to inform you that your manuscript has been judged scientifically suitable for publication and will be formally accepted for publication once it meets all outstanding technical requirements.

Kind regards,

Tunira Bhadauria, Ph.D.

Academic Editor

PLOS ONE
---

## [Editor Report · Acceptance letter]

29 Apr 2024

PONE-D-23-17536R1 

PLOS ONE

Dear Dr. Li, 

I'm pleased to inform you that your manuscript has been deemed suitable for publication in PLOS ONE. Congratulations! Your manuscript is now being handed over to our production team.

Kind regards, 

on behalf of

Dr. Tunira Bhadauria 

Academic Editor

PLOS ONE